

# Assessment of hand function in women with lymphadenopathy after radical mastectomy

Subham Mistry[1], Taimul Ali[2], Mohammed Qasheesh[3], Rashid Ali Beg[3], Mohammad Abu Shaphe[3], Fuzail Ahmad[4], Faizan Z. Kashoo[4] and Amr S. Shalaby[5]

[1] Department of Physiotherapy, KPC Medical College, Kolkata, West Bengal, India
[2] College of Physiotherapy, Peerless Hospitex Hospital & Research Center, Kolkata, West Bengal, India
[3] Department of Physical Therapy, College of Applied Medical Sciences, Jazan University, Jazan, Saudi Arabia
[4] Department of Physical Therapy & Health Rehabilitation, College of Applied Medical Science, Majmaah University, Majmaah, Saudi Arabia
[5] Faculty of Physical Therapy, Basic Science Department, Cairo University, Cairo, Egypt

Corresponding author
Mohammad Abu Shaphe,
mshaphe@jazanu.edu.sa

## ABSTRACT

**Background**. Breast cancer related upper limb lymphedema (BCRL) is a common complication in post-mastectomy patients. It is known to cause upper limb disability, which subsequently may affect the grip strength and hand function. There is little evidence on the objective assessment of functional activities particularly hand function in women with BCRL. Therefore, this study objectively assesses the handgrip strength and hand functions in women with BCRL.

**Method**. A cross-sectional study design was conducted on a sample of women with ($n = 31$) and without ($n = 31$) BCRL. The Handgrip strength and hand functions were measured using a dynamometer and Jebsen-Taylor hand function test, respectively.

**Results**. The results showed a significantly reduced handgrip strength in women with BCRL as compared to age-matched healthy adult women for both right and left hands ($p < 0.05$). Similarly, hand functions were significantly impaired in women with BCRL as compared to healthy adult women ($p < 0.05$). Reduction in handgrip strength and hand function in women with BCRL were clinically meaningful as indicated by moderate to large effect sizes (Cohen's d = 0.61 to 0.99 and 0.54 to 3.02, respectively) in all outcomes except power handgrip strength in left hand (Cohen's d = 0.38).

**Conclusion**. The results of this study indicate a significant reduction of hand grip strength and hand function in women with BCRL. Our findings suggest that objective measures of grip strength and function be included in the assessment of women with BCRL to better guide clinical decision making and patient care, which may include management of impairment associated with hand strength and function. Future studies that evaluate hand grip strength and function in a larger sample which includes a more diverse age group of women with BCRL are warranted to confirm the current findings.

## INTRODUCTION

Breast cancer related upper limb lymphedema (BCRL) is common in patients who undergo mastectomy. Lymphedema is a clinical condition in which accumulation of excess protein rich tissue fluid and tissue alterations causes oedema (*Karadibak & Yavuzsen, 2015*). The cancer registry program of twenty-five Population Based Cancer Registries (PBCRs) reports increasing trends for incidence and mortality of breast cancer in Indian women (*Malvia et al., 2017*). The cancer projection data showed that the number of breast cancer cases will increase to double by 2020 (*Malvia et al., 2017*). A previous study reported 42% incidence of lymphedema in women who underwent mastectomy (*Norman et al., 2009*).

The major signs and symptoms lymphedema are: (a) increased limb circumference, (b) restricted range of motion of affected joints, (c) stiffness, (d) sensory impairment in the hand, and (e) decrease use of affected limb for functional tasks (*Gomes et al., 2014*). Additionally, chemotherapy may cause disruption in muscle metabolism (e.g., cytokine dysregulation, adenosine triphosphate dysregulation, and deprivation of satellite cells) results muscle wasting leading to reduced muscle strength and fitness level (*Clarkson & Kaufman, 2010*).

Handgrip strength is essential for performing upper limb functional activities of daily living (*Rietman et al., 2003*). Women with BCRL may have impaired upper limb use during functional activities (*Noelle, 2005*; *Oatis, 2017*). A cross-sectional study done in 2010, compared upper extremity impairment and activity following breast cancer treatment between women with or without BCRL and reported decreased grip strength and upper limb activities in women with BCRL (*Smoot et al., 2010*). BCRL is also associated with restricted range of motion in the affected limb, reduced functional ability, as well as physical disfigurement, pain, and skin problems. There is a significant negative relationship present between severity of oedema and hand function (*Karadibak & Yavuzsen, 2015*).

In the previous study, while left and right handgrip strengths were reduced in women with BCRL as compared to age-matched healthy women, no difference between left and right handgrip strengths was found in women with BCRL (*Gomes et al., 2014*). Another study reported a significantly impaired muscle strength and function in women with breast cancer who underwent chemotherapy, or a radical mastectomy as compared to healthy women (*Klassen et al., 2017*). More recently, *Winters-Stone, Medysky & Savin (2019)* reported a significantly lower handgrip strength and function in older women with breast carcinoma than healthy older adults.

Most studies that evaluated the hand functional activities and strength in women with BCRL, have used numerous types of questionnaires such as Hand Function Sort Questionnaire (HFS) (*Karadibak & Yavuzsen, 2015*) or Disability of Arm Shoulder Hand questionnaires (DASH) (*Smoot et al., 2010*), and Hand Dynamometer (*Dawes et al., 2008*), as their outcome measures. While most studies have used subjective measures to evaluate hand function in women with BCRL, few studies have used objective measure of hand function (*Smoot et al., 2010*; *Wong et al., 2019*; *Dollar, 2014*; *Kärki et al., 2005*; *Cantarero-Villanueva et al., 2012*). For instance, *Smoot et al. (2010)* used both subjective and objective measures such as DASH questionnaire and a handheld dynamometer to evaluate hand

function and strength, respectively in women with or without BCRL. *Cantarero-Villanueva et al. (2012)* examined handgrip strength as an objective measure of function in breast cancer survivors. It is important to objectively assess hand grip strength and hand function so that appropriate and complete clinical assessment and intervention can be implemented to address any impairments or functional limitations in women with BCRL (*Mak et al., 2015*; *Rietman et al., 2004*; *Hayes et al., 2010*; *Park, Jang & Seo, 2012*).

There is limited research to determine objective assessment of hand function after surgery and chemotherapy treatment in women with breast cancer. Therefore, the purpose of this study is to (1) evaluate the hand grip strength and hand functional activities using objective outcome measures in women with BCRL; and (2) to compare findings in women with BCRL to healthy control. In women with BCRL, objective assessment of hand function will provide greater clarity and precision regarding performance of functional activities of daily living than subjective assessment alone, which will better guide clinical practice. The current study hypothesizes that grip strength and objectively measured hand function will be impaired in women with BCRL compared to women without BCRL.

## MATERIALS & METHODS

This study was an observational cross-sectional design with convenience sampling. All the subjects were recruited from Ramaiah Medical College Hospital and HCG MSR Cancer Centre, Bengaluru, India. An ethical clearance was obtained from the Ethical Committee of Ramaiah Medical College and Hospital (MEU-PT/EC/12/2018). Purpose of the study was explained to each individual and a written informed consent was obtained.

The subjects were included based on the following criteria: (1) age between 35–65 years, (2) had completed chemotherapy/surgery >6 months, (3) women with BCRL for more than 3 months old, (4) affecting the dominant hand and (5) self-reported pain score of seven or less on visual analogue scale (because >7 pain score might influence handgrip strength and function) (*Cantarero-Villanueva et al., 2012*). Subjects were excluded if they were receiving radiotherapy, who had a history of fracture or surgery in upper limb in last 3 months or had neuromuscular or musculoskeletal disorders that would have prevented assessment of hand grip strength and function. The control group included a convenience sample of 31 healthy age-matched women without history of breast carcinoma.

### Objective measures

A hand dynamometer and pinch gauge/pinch meter were used to measure power grip and precision grip strength and the Jebsen-Taylor hand function test (JTHFT) (*Mak et al., 2015*) tool kit was used to assess hand function. Hand dynamometer and pinch gauge are valid instruments to assess grip strength (*Neumann et al., 2017*; *Lindstrom-Hazel, Kratt & Bix, 2009*; *Shin et al., 2012*). Handgrip strength and function of both patient (i.e., women with BCRL) and healthy control (i.e., women without BCRL) samples were evaluated in a similar fashion.

### Measurement of power grip strength of hand

A baseline hand dynamometer was used to assess the power grip. Individuals were asked to hold the hand dynamometer, where the elbow was in 90 degrees flexion and shoulder

in neutral position. The individuals were asked to press the hand dynamometer as hard as they can, three times, and the best value was taken from each hand (*Smoot et al., 2010*).

## Measurement of precision grip strength of hand

For assessing the precision grip, pinch gauge was used (*Smoot et al., 2010*). Individuals were asked to hold the pinch gauge and press it in three different position, Tip of thumb to tip of index finger, thumb pad to lateral aspect of index finger, and thumb pad to pad of index finger and pad of middle finger (Tripod pinch). Individuals sat on a chair, their elbow was in 90 degrees flexion and shoulder in neutral position. Instructions were given to the patients to press the pinch gauge in three different position (Tip to tip, Key pinch, and Tripod pinch) as hard as they can. Each precision grip was performed three times and the best value was taken from each hand.

## Assessment of hand function

For assessing bilateral hand function, the Jebsen-Taylor hand function test (JTHFT) (*Mak et al., 2015*) tool was used. It is a valid and reliable tool which objectively measures the hand function (*Mak et al., 2015*). It is a task specific tool, consisting of 7 tasks for assessing hand functions. The tasks are: writing, picking up small common objects, and picking up large objects, card turning, simulated feeding, and stacking checkers. Each task was measured by calculating time (*Mak et al., 2015*).

## Statistical analysis

The data were tabulated in Microsoft Excel and the statistical program for social science software for Window (version 17, SPSS Inc, Chicago, IL) was used for the statistical analyses. Descriptive statistics was used to calculate the mean and standard deviation of the patient's age. A student $t$-test was used to determine statistical difference of grip strength and hand function between normative data and patient data. Additionally, the effect sizes were calculated using the Cohen's d for each variable to evaluate clinically meaningful changes. Effect size were defined as: small ($d =< 0.5$), medium ($d = 0.50$ to $0.80$), and large ($d => 0.80$) (*Goulet-Pelletier & Cousineau, 2018*). Results were considered statistically significant if $p < 0.05$. The sample size was calculated using estimation of means from the formula ($n = [Z\alpha\sigma/d]^2$). Where Z $\alpha$ is 95% confidence level, $\sigma$ is standard deviation, and d is the margin of error. The Z $\alpha$ was 1.96, the Standard deviation was 2.1 of writing in JTHFT and the margin of error was 0.75 (*Gärtner et al., 2010*). The estimated sample size came to 31.

## RESULTS

A total of 31 women with BCRL and 31 age-matched healthy women were included in the study (Table 1). Out of 31 subjects, 21 subjects had right hand lymphedema and 10 subjects had left hand lymphedema. Sixty-eight percent women with BCRL and 58% healthy women were right hand dominant. Mean age of the patients and healthy control groups were 55.5 ($\pm$ 8.4) and 55.7 ($\pm$ 8.1) years, respectively.

Table 2 presents the result of power grip and precision grip strength of right and left hands. Compared to healthy controls, there were significant reductions of both power and

**Table 1  Participants characteristics.**

| Variables | Women with BCRL Mean ± SD | Women without BCRL Mean ± SD | P (t) |
|---|---|---|---|
| Age (years) | 55.5 ± 8.4 | 55.7 ± 8.1 | 0.890 (0.138) |
| Body mass index (kg/m$^2$) | 24.5 ± 0.7 | 24.4 ± 0.8 | 0.842 (0.201) |
| Hand dominance (% of right) | 68% | 58% | 0.439 (0.779) |
| Side of BCRL (right/left) | 21/10 | | |
| Duration of BCRL (months) | 8.7 ± 3.4 | | |
| Treatment of cancer | | | |
| Surgery | 11 | | |
| Surgery and chemotherapy | 11 | | |
| Surgery and radiotherapy | 9 | | |
| Number of nodes removed | 10.4 ± 2.1 | | |

**Notes.**
BCRL, Breast Cancer related Lymphedema.

precision grip strengths $p < 0.05$. Reductions in handgrip strength in women with BCRL were clinically meaningful as indicated by moderate to large effect sizes (Cohen's $d = 0.61$ to 0.99) in all handgrip strength except power handgrip strength in left hand (Cohen's $d = 0.38$).

Table 3 compares the activities of hand functions in women with and without BCRL. The hand functions were significantly reduced across all activities in women with BCRL when compared to the healthy controls ($p < 0.05$). Reduction in hand function in women with BCRL were clinically meaningful as indicated by moderate to large effect sizes (Cohen's $d = 0.54$ to 3.02, respectively) in all variables of hand functions.

## DISCUSSION

The present study examined the hand grip strength and hand functional activities in women with BCRL. Findings of the study indicate significant reduction in the power grip and precision grip strengths in women with BCRL as compared to age-matched healthy adult women. The results of the hand function also showed significantly increased time to complete the task performance. Reduction in handgrip strength and hand function in women with BCRL were also clinically meaningful as determined by moderate to large effect sizes (Cohen's $d = 0.61$ to 0.99 and 0.54 to 3.02, respectively) in all outcomes except power handgrip strength in left hand (Cohen's $d = 0.38$).

Reduced strength has been known to have debilitating effects on women with BCRL. Most studies demonstrate that upper limb lymphedema negatively affects the hand grip strength (*Wong et al., 2019*). The same was found to be true among women with BCRL in the current study, mean of the dominant right-hand power grip was 23.8 kg and dominant left hand was 22.2 kg when compared to the control group of 29.4 and 25.5, respectively. The reduction of grip strength could be due to swelling in hand and wrist which in turn leads to decreased wrist and finger range of motion (ROM) and reduction of initiation of wrist extension and finger flexion (*Smoot et al., 2010*). The position of producing a power
**Table 2** Comparison of grip strength between women with and without breast cancer related lymphedema.

| Variables | RIGHT | | | | LEFT | | | |
|---|---|---|---|---|---|---|---|---|
| | Women with BCRL Grip strength (Kg.) Mean ± SD | Women without BCRL Grip strength (Kg.) Mean ± SD | p value (<0.05) | Effect size (Cohen's d) | Women with BCRL Strength Mean ± SD | Women without BCRL Grip strength (Kg.) Mean ± SD | p value (<0.05) | Effect size (Cohen's d) |
| Power grip | 23.8 ± 9.9 | 29.4 ± 8.3 | 0.019 | 0.61 | 22.2 ± 9.3 | 25.5 ± 7.9 | 0.013 | 0.38 |
| Precision grip | | | | | | | | |
| Tip pinch | 4.1 ± 1.1 | 5.2 ± 1.7 | 0.004 | 0.76 | 3.9 ± 1.1 | 5.0 ± 1.7 | 0.004 | 0.77 |
| Key pinch | 6.1 ± 1.3 | 7.5 ± 2.6 | 0.013 | 0.68 | 5.9 ± 1.3 | 7.3 ± 2.9 | 0.033 | 0.62 |
| Tripod pinch | 5.4 ± 1.1 | 7.5 ± 2.8 | 0.001 | 0.98 | 5.2 ± 1.1 | 7.3 ± 2.8 | 0.001 | 0.99 |

**Notes.**

BCRL, Breast Cancer related Lymphedema; Effect size, small (0.20), medium (0.50), large (0.80).

Mistry et al. (2021), *PeerJ*, DOI 10.7717/peerj.11252

**Table 3  Comparison of hand functions between women with and without breast cancer related lymphedema.**

| Variables | RIGHT | | | | LEFT | | | |
|---|---|---|---|---|---|---|---|---|
| | Women with BCRL Hand function (Sec) Mean ± SD | Women without BCRL Hand function (Sec) Mean ± SD | *p* value (<0.05) | Effect size (Cohen's d) | Women with BCRL Hand function (Sec) Mean ± SD | Women without BCRL Hand function (Sec) Mean ± SD | *p* value (<0.05) | Effect size (Cohen's d) |
| Card turning | 7.5 ± 2.5 | 4.3 ± 1.6 | 0.010 | 1.53 | 8.2 ± 2.7 | 4.9 ± 1.4 | 0.001 | 1.53 |
| Picking up small objects | 7.1 ± 1.7 | 4.9 ± 1.1 | 0.002 | 1.54 | 7.4 ± 2.1 | 5.2 ± 1.5 | 0.021 | 1.21 |
| Simulated feeding | 8.4 ± 1.4 | 5.1 ± 0.8 | 0.003 | 2.89 | 9.7 ± 1.4 | 5.9 ± 1.1 | 0.003 | 3.02 |
| Stacking checkers | 3.8 ± 0.9 | 3.3 ± 0.6 | 0.035 | 0.65 | 4.1 ± 1.4 | 3.5 ± 0.7 | 0.001 | 0.54 |
| Picking up large objects | 4.3 ± 1.0 | 2.9 ± 0.6 | 0.001 | 1.70 | 5.3 ± 2.5 | 3.1 ± 1.4 | 0.001 | 1.09 |
| Picking up heavy objects | 4.3 ± 0.9 | 2.9 ± 0.6 | 0.023 | 1.83 | 5.2 ± 1.9 | 3.2 ± 1.1 | 0.001 | 1.29 |
| Handwriting | 33.2 ± 26.4 | 21.2 ± 12.2 | 0.025 | 0.58 | 64.5 ± 32.1 | 39.8 ± 10.1 | 0.001 | 1.04 |

**Notes.**

BCRL, Breast Cancer related Lymphedema; Effect size, small (0.20), medium (0.50), large (0.80).

grip for normal individual is, wrist in slight extension and elbow in 90 degrees of flexion, in BCRL patients the wrist extension is reduced due to swelling resulting in active insufficiency of hand muscles, which could reduce the strength of grip. A study conducted by *Dawes et al. (2008)* found similar results where women with BCRL had lesser hand grip strength, and shoulder ROM and other hand morbidities, which led to impairment. In precision grip strength, the activity of tip, key and tripod pinch requires much finer motor control and is more dependent on intact sensation. In case of women with BCRL, it is also seen that there is involvement of kinesthetic sense of wrist joint and small joints which reduces the force generation during gripping activities (*Karadibak & Yavuzsen, 2015*; *Smoot et al., 2010*), this could be the probable reason for the reduced precision grip strength in the studied subjects.

BCRL and its complications involving impaired upper limb function are well established sequelae among breast cancer survivors. About 13–28% of breast cancer survivors post treatment when surveyed have reported limitations in daily activities (*Voogd et al., 2003*). The present participants when objectively assessed (JTHFT) for their hand function showed similar results to that available in literature. For the JTHFT when administered to the women with BCRL, the duration of each activity like writing, picking up small common objects, and picking up large objects, card turning, simulated feeding, and stacking checkers performance was increased. The overall time taken for completion of tasks was 68.8 s for right hand dominant and 100.3 s for left hand dominant.

While performing JTHFT, the tasks require power grip and precision grip equally, like picking up large and heavy objects, handwriting, simulated feeding etc. Reduced power grip strength and precision grip strength will have implications on activity performance. Our subjects also showed reduced strength and task time was increased, thereby we can assume that strength does play a role in performing of hand functions. A study conducted by *Dawes et al. (2008)* reported that when DASH questionnaire along with grip strength assessment of women with BCRL was analyzed, the scores were higher in women who had self-reported symptoms of lymphedema, indicating activity limitation and participatory restriction. This reduction of hand function could be because of the reduction in power grip strength and precision grip strength and reduction of kinesthetic sense (*Karadibak & Yavuzsen, 2015*), thus suggesting that impairment in hand grip strength plays a part in performing hand function. Loss of muscle strength and lack of movement in articulation could also add to the reasons for impaired hand function (*Karadibak & Yavuzsen, 2015*; *Rietman et al., 2003*; *Smoot et al., 2010*; *Dawes et al., 2008*; *Kärki et al., 2005*).

BCRL or hand oedema significantly affects functional mobility and activities of daily livings of upper extremity (*Karadibak & Yavuzsen, 2015*; *Rietman et al., 2003*; *Smoot et al., 2010*; *Dawes et al., 2008*). Additionally, women with BCRL may develop weaknesses and restriction of shoulder muscles and range of motion and reduced quality of life (*Karadibak & Yavuzsen, 2015*). *Voogd et al. (2003)* reported reduced daily functional skills, lack of energy and motivation, and reduced quality of life.

### Limitations and directions for future research

The current study acknowledged some limitations. First, this study used a cross-sectional design, therefore, a causal relationship between BCRL, and hand grip strength and function could not be established. Future prospective longitudinal studies that evaluate hand function during and after rehabilitation of women with BCRL are warranted to examine changes in hand grip strength and function as compare to age matched healthy individuals. Second, the results of this study were limited to the specific age groups (i.e., 35 to 65 years), hence generalization of this results in younger (e.g., <35 years) or older (e.g., >65 years) women with BCRL need caution. Further studies that included a more diverse age group (e.g., young to elderly) are required to examine influence of age on hand grip strength and function in women with BCRL. Third, although the current study estimated a priori sample size, larger sample might give better results.

## CONCLUSIONS

The results of this study indicate a significant reduction of hand grip strength and hand function in women with BCRL. Our findings suggest that objective measures of grip strength and function be included in the assessment of women with BCRL to better guide clinical decision making and patient care, which may include management of impairment associated with hand strength and function. Future studies that evaluate hand grip strength and function in a larger sample with a more diverse age group of women with BCRL population are warranted to confirm the current findings.

### Funding
The authors received no funding for this work.

### Competing Interests
The authors declare there are no competing interests.

### Author Contributions
- Subham Mistry and Taimul Ali conceived and designed the experiments, performed the experiments, prepared figures and/or tables, authored or reviewed drafts of the paper, and approved the final draft.
- Mohammed Qasheesh, Fuzail Ahmad and Faizan Z. Kashoo conceived and designed the experiments, analyzed the data, authored or reviewed drafts of the paper, and approved the final draft.
- Rashid Ali Beg conceived and designed the experiments, prepared figures and/or tables, authored or reviewed drafts of the paper, and approved the final draft.
- Mohammad Abu Shaphe conceived and designed the experiments, performed the experiments, analyzed the data, prepared figures and/or tables, authored or reviewed drafts of the paper, and approved the final draft.

- Amr S. Shalaby conceived and designed the experiments, authored or reviewed drafts of the paper, and approved the final draft.

## Human Ethics

The following information was supplied relating to ethical approvals (i.e., approving body and any reference numbers):

Ethical approval was obtained from the Ethical Committee of Ramaiah Medical College and Hospital (MEU-PT/EC/12/2018).

## Data Availability

The raw data are available in the Supplemental File.

## Supplemental Information

Supplemental information for this article can be found online at http://dx.doi.org/10.7717/peerj.11252#supplemental-information.

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
