# Peer review of "Assessment of hand function in women with lymphadenopathy after radical mastectomy"

_PeerJ, doi:10.7717/peerj.11252_

## Round 0.1 · original submission · Major Revisions

Dr Shaphe and co-authors, your manuscript "Assessment of hand function in women with lymphadenopathy after radical mastectomy" has been reviewed and the Reviewers have recommended you address a number of issues they have outlined in their reviews. Thank you for your submission to PeerJ and I look forward to receiving your Responses to Reviewers and amended manuscript in a timely manner. Thanks, Assoc Pprof Mike Climstein (FASMF, FACSM. FAAESS)

Reviewer 1 ·

Basic reporting

Thank you for conducting this study in the much needed area of function in women with breast cancer related lymphedema. I hope that you find the comments and suggestions helpful.

There are some opportunities for improvement in flow and meaning in the introduction and methods, and correction of grammatical errors and use of punctuation here and there.

Introduction

1. Line 37: No comma after patients.

2. Lines 39-41: This sentence could be much clearer. I suggest (if this still accurately reflects your meaning): The cancer registry program of twenty-five Population Based Cancer Registries (PBCRs) reports increasing trends for incidence and mortality of breast cancer in Indian women.

3. Lines 55-56: You wrote: “It is also seen that lymphedema cause restricted range of motion in the affected limb, reduced functional ability as well as physical disfigurement, pain, and skin problems.” I suggest instead: “BCRL is also associated with restricted range of motion in the affected limb, reduced functional ability, as well as physical disfigurement, pain, and skin problems.”

4. Line 73: change “done to assess” To “that evaluated”

5. Use person-first language: Instead of “lymphedema patients”, say “in women with BCRL”

6. Once you use the acronym BCRL that first time, you should use that throughout… you don’t need to spell it out anymore, nor do you need to say lymphedema when referring to breast cancer related lymphedema; you can simply use BRCL.

7. Try to be more consistent with the use of patients, women, participants, and subjects. Where appropriate, use a consistent term for the women enrolled in your study.

8. Line 62-64: You wrote: “While most of the studies subjectively measure the hand function, there are few studies available that have measured objectively the hand function in patients with breast cancer related upper limb lymphedema. I suggest a slight edit to: “While most studies have used subjective measures to evaluate hand function in women with BCRL, few studies have used objective measure of hand function.” I suggest moving the information about the reliability and validity of the tools used in this study to the methods section.

9. And then move lines 69-71 to the end of what is currently line 64. I also suggest these edits to that sentence: You wrote: “It is important to assess hand grip strength and hand function, so that appropriate clinical assessment and intervention planning of lymphedema can be based on it.” I suggest: “It is important to objectively assess hand grip strength and hand function so that appropriate and complete clinical assessment and intervention can be implemented to address any impairments or functional limitations”.

10. Lines 72-76: You wrote: “Objective assessment of hand function will give more clarity on the level of independence in performing functional activities of daily living comparatively better than subjective assessment of hand function, hence there is requirement of assessing hand functional activity objectively in patients with BCRL. This study evaluated the hand grip strength and hand functional activities in people with BCRL”. I suggest modifying this last paragraph to explicitly state the study purpose or aim. You also need to add that you’re comparing your findings to normative data. My suggestions: The purpose of this study is to 1) evaluate the hand grip strength and hand functional activities using objective outcome measures in women with BCRL; and 2) to compare findings in women with BCRL to normative data. In women with BCRL, objective assessment of hand function will provide greater clarity and precision regarding performance of functional activities of daily living than will subjective assessment alone, which will better guide clinical practice.

11. I also highly recommend you include a hypothesis statement in that last paragraph if you have one. The study purpose and hypothesis are not clearly stated. I’ve made some recommendations with reference to that. In addition, the current introduction does not mention normative data or comparison to that (which should be done in the purpose statement). A hypothesis could be stated as such: “We hypothesize that grip strength and objectively measured hand function will be impaired in women with BCRL compared to normative data.”

Materials and Methods

1. Line 78: Change “It” to “This study”

2. Move the information regarding sample size to the Statistical Analysis section at the end of methods.

3. The word data is plural and should be followed by were or are, not was or is.

4. Add the information about reliability and validity of the outcomes to the appropriate section in methods.

5. Statistical Analysis section: Move sample size estimate here.

Literature references, sufficient field background/context

1. The article includes sufficient introduction to breast cancer related lymphedema. There are some areas that need to be developed and some references need to be checked and potentially updated.

2. Lines 42-43: You should use primary references here. The reference you cite is for a different study – not an incidence study.

3. Lines 44-49: The reference you cite for this paragraph supports only the first sentence of the paragraph. While the authors of the paper you cite do describe the finding in your second sentence, they are citing a different study. It would be best to cite the original, primary, source for your references whenever possible.

4. Line 51: You refer to 25% incidence of BCRL here, but earlier you say 5.5 to 56% incidence. I suggest being consistent, or you could modify this sentence to say: “Women with BCRL may have impaired upper limb use during functional activities.” (and, of course, keep the citation)

5. There are two issues you need to address in terms of background and gaps in the literature:

You don’t talk about what normal grip strength or hand function is. You compare your data to normative data but never talk about the normative studies. You should add a discussion of what normal strength is in women in this age group (you can cite the studies that use the normative data you’ll be comparing too). Also relate those findings to the findings from other papers. How much have the other studies found grip strength to be impaired in BCRL cases vs non-cases, or compared to normative data? Consider adding this after line 58.

The paper would really benefit from a more detailed discussion of the BCRL literature on hand function to address these questions: Of the studies that looks at patient reported outcomes of function, what did they show? Of the studies that did look at objective measures of function, what did they use, and what did they show? You write that “there are few studies available that have measured objectively the hand function in patients with breast cancer related upper limb lymphedema.” (line 63). Few implies that some were done. If none were done, say “No studies have been identified that”. If there are some, talk about their findings. The you can come back to that in your discussion to compare and contrast your findings to theirs.

6. At your current line 71, you have 4 references. I’m not clear about the relevance of reference #16 here. (Rhee H, Yu J, Kim S. Influence of Compression Types on Hand Function: A Preliminary Investigation. Journal of Physical Therapy Science. 2011;23(3):477–80.)

Article structure, figures, tables:

The structure of the article conforms to the acceptable journal section format. Two tables are included. I suggest the addition of a Participant Characteristic Table (as table 1 with changes to titles to Table and 3 as a result) with age, body mass index, hand dominance, side of BCRL, type of cancer treatment (surgery, radiation, therapy, chemotherapy), number of nodes removed, duration of BCRL, etc.). This is important so that the reader knows who the participants were. The reader cannot generalize the results of this study to their own patients if they don’t know how similar they are. This speaks to the external validity of the study.

All appropriate raw (outcomes) data has been made available.

Experimental design

The paper fairly clearly defines a gap that the authors are trying to fill – the lack of assessment of hand function and grip strength using objective data. Grip strength is most often assessed with objective measures (dynamometry) and has been reported in a number of studies. However, the assessment of hand function using an objective tool is novel and has clinical value. However, there are some additional gaps that I’ve described above.

2. Line 90: Participant eligibility: In exclusion criteria, what is meant by “recent” in terms of fracture or surgery?

3. Lines 91-92. Delete and add below a subheading. My suggestion:

Objective Measures

A hand dynamometer and pinch gauge/ pinch meter were used to measure power grip and precision grip strength and The Jebsen-Taylor hand function test (JTHFT) Tool Kit was used to assess hand function. Hand dynamometer and pinch gauge are valid instruments to assess grip strength. (And add a citation for this last sentence)

This study is a single group cross sectional study, comparing findings in women with BCRL to normative data. The biggest issue that must be addresses is that there is no indication of how the normative data was derived, who they are, their age, etc. The comparison is not valid without details about the sample from which this the normative data arose.

in the intro and methods, a discussion of normative data may need to be added. Without knowledge of the normative data, the conclusions cannot be confirmed.

Validity of the findings

The outcome data looks good. However, there are some threats to study validity that I hope can be addressed.

Internal validity is threatened by 1) unknown source of normative data, and 2) potential for confounding effects of unknown clinical and demographic characteristics, which were not reported.

External validity is threatened by sampling but more so by the fact that the sample is poorly described. In cross sectional designs, internal validity can be improved by attempting to control for confounders during the sampling process (inclusion and exclusion criteria). While external validity may be sacrificed for these tighter inclusion criteria, at least the study conclusions will be more trustworthy. However, we don’t know much about the sample to see if there is an underlying issue with potential confounders. If all the women had chemotherapy induced neuropathy, for example, we can’t be sure that the association found (between BCRL and hand strength/function) is a true associations.

The data for the sample studied has been provided. Thank you. However, references for the normative data were not provided.

Clinical and demographic characteristics are poorly reported and are not controlled for.
In the discussion (lines 153-155), you state: “The overall time taken by the subjects for completion of tasks was 68.8 seconds for right hand dominant and 100.3 seconds for left hand dominant.” Were all your participants right hand dominant? If so, you need to very clearly state this where ever it’s needed. Dominance is a potential confounder, and if you controlled for that during your sample selection, include that in your paper please (e.g., as an inclusion criteria). You do say that the BCRL has to be the dominant side, but not that it had to be right or left sided. According to your data spreadsheet, you have both right and left affected (thus dominant) participants.

The conclusion should be modified slightly. You state (line 174-177): The results of this study indicate a significant reduction of hand grip strength and hand function, resulting in reduction in functional activities in patients with BCRL. These impairments are amenable following an upper limb rehabilitation intervention. Based on the result of this study, physical rehabilitation is recommended in the management of BCRL.” There are two concerns here. 1) You can’t say that the reduction in strength and function result in reduction in functional activities. You didn’t test that. I suggest removing “resulting in reduction in functional activities in patients with BCRL” from the sentence. 2) You also can’t say that the impairments are amenable to intervention and that based on your findings rehab is recommended because you didn’t test that either. You can say that based on your findings you recommend that objective measures of grip strength and function be included in the assessment of women with BCRL to better guide clinical decision making and patient care, which may include impairment associated with hand strength and function. You can use words like “our findings suggest”, or “we speculate that”…. Etc.

In addition, your paper needs a limitation sections, and directions for future research.

Reviewer 2 ·

Basic reporting

Article is written with clear and concise English. There were places where references were missing and those places are highlighted. A comment was made on the figure.

Experimental design

The research question is clear. The authors did a nice job with identifying the knowledge gap. A few comments are made to improve the methods section.

Validity of the findings

The study uses valid and reliable instruments for the assessments. The sample size is small and seems like a pilot study. This can be highlighted in the conclusion. Future studies would need to be done on a bigger sample to confirm findings. The data analysis is done appropriately and I have made a few suggestions for improvement.

Additional comments

This important study should be published with minor revisions as suggested.

Annotated reviews are not available for download in order to protect the identity of reviewers who chose to remain anonymous.

---

## Round 0.2 · Minor Revisions

Dr Shaphe and co-authors. Thank you for the changes to your manuscript which the Reviewers have noted. One Reviewer would like minor/further changes which they have indicated on the attached PDF. Please ensure you make all of the changes and the Reviewer has recommended that you use the term "normative data". They further recommend a switch to healthy controls or control group data.

I look forward to receiving your amended manuscript. Thanks, A/Prof Mike Climstein

Reviewer 1 ·

Basic reporting

Thank you for your attention to the suggested revisions! The reporting is clear, the appropriate background literature has been discussed and references, the results and conclusions are consistent with the hypothesis. I like the addition of effect sizes as suggested by the other reviewer.

Experimental design

The hypothesis and study purpose are present and clear. The gap in the literature has now been addressed with the modifications. Methods are clear.

Validity of the findings

Data has been provided. Analysis is clear. Conclusions are valid and not over-reaching.

Additional comments

The revisions were acceptable and well done. I have one suggestion - rather than use the term "normative data" switch to "control group data" or healthy controls. That's actually a more accurate reflection of what you did. You didn't compare your BCRL group to "normative data" per se. I tried to correct that in the annotated pdf.

Annotated reviews are not available for download in order to protect the identity of reviewers who chose to remain anonymous.

---

## Round 0.3 · accepted · Accept

Dear Assoc. Professor Shape, thank you for your timely changes to your manuscript as recommended by the Reviewers "Assessment of hand function in women with lymphadenopathy after radical mastectomy". I am pleased to inform you that I am recommended your manuscript for publication in PeerJ.

Thank you and your colleagues for supporting PeerJ and we look forward to future submissions. Congratulations on your publication, A/Prof. Mike Climstein